# Wearable Electronic Tongue for Non-Invasive Assessment of Human Sweat

**DOI:** 10.3390/s21217311

**Published:** 2021-11-03

**Authors:** Magnus Falk, Emelie J. Nilsson, Stefan Cirovic, Bogdan Tudosoiu, Sergey Shleev

**Affiliations:** 1Department of Biomedical Science, Faculty of Health and Society, Malmö University, 205 06 Malmö, Sweden; magnus.falk@mau.se (M.F.); emelie.nilsson@mau.se (E.J.N.); stefan.cirovic@mau.se (S.C.); 2Biofilms—Research Center for Biointerfaces (BRCB), Malmö University, 205 06 Malmö, Sweden; 3Covercast AB, Drottensgatan 4, 222 23 Lund, Sweden; bogdantds@gmail.com

**Keywords:** electronic tongue, human sweat, non-invasive analysis, wearable sensors

## Abstract

Sweat is a promising biofluid in allowing for non-invasive sampling. Here, we investigate the use of a voltammetric electronic tongue, combining different metal electrodes, for the purpose of non-invasive sample assessment, specifically focusing on sweat. A wearable electronic tongue is presented by incorporating metal electrodes on a flexible circuit board and used to non-invasively monitor sweat on the body. The data obtained from the measurements were treated by multivariate data processing. Using principal component analysis to analyze the data collected by the wearable electronic tongue enabled differentiation of sweat samples of different chemical composition, and when combined with ^1^H-NMR sample differentiation could be attributed to changing analyte concentrations.

## 1. Introduction

A significant effort is being devoted to the development of new healthcare and fitness innovations, driven by the need imposed by patients and individuals, as well as the possibilities provided by recent electronics, to evaluate and benchmark personal health parameters [1,2]. By designing wearable devices, it is possible to perform continuous, non-invasive monitoring of biomarkers for assessing human performance, health, and wellbeing. Such new wearable technologies enable a shift from professional medical care provided by hospitals to essentially outsourced medical services at point-of-care units and homes and individuals being able to self-assess, at a much lower cost than what is currently possible. A variety of different wearable electronics systems are already on the market, which are capable of, e.g., monitoring heart rate and maximal oxygen uptake (VO2 max). However, these systems measure general parameters, and do not perform non-invasive chemical analysis.

To facilitate non-invasive sensing, there is a large interest in developing different sensors and biosensors operating in interstitial fluid, saliva, tears, and sweat, which promises continuous analyte access and measurement in a minimally- or non-invasive format [3]. In particular, there is a growing interest to use perspiration for chemical analysis as sweat contains a wealth of physiologically and metabolically important biomarkers, such as electrolytes and metal ions, metabolites, amino acids, proteins, and hormones, and can be sampled non-invasively [4,5,6,7,8]. Traditionally, the high demands on laboratory infrastructure, typically needing mass spectrometry or H-NMR to analyze the samples, have thus far prevented the clinical implementation of sweat as a diagnostic biofluid, limiting the clinical importance of sweat to the determination of chloride for the diagnosis of cystic fibrosis [9,10]. The content of sweat is also highly variable, both between individuals but also dependent on the sample location from the same individual, making sample analysis harder [11].

The use of sweat as a noninvasive, laboratory-independent, on-skin diagnostic biofluid has the potential to have a major impact on health care in the future. Sweat analysis is an ideal method for continuously tracking a person’s physiological state, but no commercial portable device exists as of yet whereas several different wearable sweat monitoring devices have been reported recently by different research groups [3,12,13,14,15]. A sensitive and selective approach to multiplexed sensing of sweat biomolecules can be achieved by utilizing wearable electrochemical sensors, coupled with electrical components for signal transduction and data transmission. One such example was described by Gao et al., where the authors designed a sensor array device of different selective electrodes, combining biosensors for detection of glucose and lactate with ion-selective electrodes for sodium and potassium [12]. By having a sensor array where the individual sensors showed high specificity towards a different specific analyte, the authors could measure a detailed sweat profile of human subjects. However, it is a well-known fact that biosensors have quite restricted operational lifetime due to the limited stability of enzymes, and most of them are not ready for real practical usage outside scientific, industrial, or medical laboratories.

An alternative approach to using highly specific electrodes for sweat analysis is instead to employ so called electronic tongues (e-tongues), composed of an array of robust and stable but non-specific, poorly selective sensors with partial specificity and using data-processing algorithms, such as principal component analysis (PCA) and hierarchical cluster analysis (HCA), to analyze the sample [16]. In addition, sweat is a complex solution containing a mixture of many electrochemically active high and low molecular weight compounds, which complicates electrochemical detection of individual compounds. In an e-tongue, the performance of individual sensors can be greatly improved in terms of the limit of detection and selectivity by the simple inclusion of data from seemingly non-related sensors [17,18]. The capabilities of e-tongue systems show a unique ability to deal with complex and changing background and diminish the impact of interferents, and a wide variety of voltammetric e-tongues have been used, e.g., analysis of foodstuff or environmental monitoring [19,20,21,22,23,24,25,26,27]. Recently, a portable electronic tongue was developed by a research team from IBM for classification of different beverages [28]. The device was connected to cloud computing and employed machine learning algorithms for the classification of liquids within less than one minute. E-tongue systems can also be used as a diagnostic approach to identify and monitor early stages of pathological biological processes in complex biological fluids, and have been used to investigate urine samples and for continuous monitoring in saliva [29,30,31,32]. A potentiometric ion-selective sensor array has also been used to classify cystic fibrosis from stimulated sweat samples [33]. However, wearable e-tongues have so far not been used for the analysis of human perspiration.

The aim of this work was to investigate the use of a voltametric e-tongue, composed of different metal electrodes, for the differentiation of sweat samples containing different analytes of interest, employing mainly PCA to differentiate the samples. Based on the results from the analysis of the e-tongue in sweat samples in vitro, a miniature wearable voltametric tongue, where metal electrodes were incorporated on a flexible circuit board, was designed and investigated for on-body measurements of human perspiration, where the content of the characterized sweat samples also were analyzed with ^1^H-NMR, combining the non-specific response of the e-tongue with analytical determination of the fluid to assess the compositional change of the sweat, leading to differentiation in the sample-classification by the e-tongue.

## 2. Materials and Methods

### 2.1. Chemicals and Materials

In total, three different e-tongues were used for the investigations. For the initial characterization of the e-tongue in buffer and sweat, macro electrodes from BASi^®^ (Bioanalytical Systems Inc, West Lafayette, IN, USA) made of gold and platinum (2 mm in diameter) and palladium (3 mm in diameter) were used (Device 1). To be able to investigate small volumes of sweat, a micro e-tongue was designed, made from metal wires from Goodfellow Cambridge Ltd. (Huntingdon, UK) of palladium (0.125 mm in diameter), gold and platinum (0.1 mm in diameter), all with a length of 1.5 cm of the working electrode, where the remaining wire was insulated using cellulose acetate dissolved in acetone with a concentration of 15 mg mL^−1^ (Device 2). A few mLs of sweat was used for each measurement. Finally, a wearable e-tongue prototype was designed by plating gold, palladium, and platinum as working electrodes (1 mm in diameter) and silver (3 mm in diameter) as a combined counter/reference on a flexible circuit board (Device 3). The metal electrodes were electroplated onto conducting copper tracks at SIFCO ASC Sweden AB (Sifco Asc Sweden AB, Rättvik, Sweden). A nine times larger area was chosen for the counter/reference electrode compared to the working electrode area to minimize deviations in the reference potential [34]. The circuit board was covered with neoprene rubber, leaving channels where the sweat could flow to the electrodes. The neoprene rubber was coated with S100 hydrophilic coating (Jonsman Innovation ApS, Gørløse, Denamrk) to facilitate the flow of sweat to and from the electrodes.

All chemicals used were of analytical grade, obtained from Sigma Aldrich (St. Louis, MO, USA). The buffer used was a 10 mM phosphate buffer (PB), pH 6.9 containing 10 mM NaCl. For the initial characterization of the electronic tongue, the PB was spiked with different analyte concentrations: 150 mM sodium chloride (NaCl), 5 mM glucose (Glc), 10 mM lactate (Lac), 100 µg mL^−1^ ascorbic acid (AA), 10 mM urea (Ur), 100 mM sodium bicarbonate (NaCarb), and 10 mg mL^−1^ albumin (Alb), respectively. For characterization of sweat in vitro, a single sample was collected from a healthy volunteer and used for characterization of the same analytes as with PB. The pH value of the PB was chosen to match the measured pH value of this sweat sample. Additional sweat samples were collected from three healthy volunteers and analyzed separately. All sweat samples used for in vitro analysis were collected via heat-induced sweating, pooled from collected sweat from the forehead, back, chest, and arms.

### 2.2. Electrochemical Measurements

Electrochemical characterization was performed using a DropSens multichannel Potentiostat µStat 8000P (Oviedo, Spain). For characterization of Device 1 a standard three electrode configuration was used with an Ag/AgCl (sat.) reference electrode and a platinum counter electrode. For analysis of sweat with the micro e-tongue (Device 2) and wearable e-tongue (Device 3), a 3 cm silver wire (0.125 mm in diameter, Goodfellow Cambridge Ltd., Huntingdon, England) and a plated silver electrode were used as a combined counter/pseudoreference Ag/AgCl electrode, respectively. Prior to measurement, the electrodes were gently polished with 0.05 μm aluminum oxide powder from Struers (Westlake, OH, USA) and thereafter cleaned electrochemically by cycling in 0.5 M H_2_SO_4_.Fluid samples were characterized using differential pulse voltammetry (DPV). A potential range from −0.6 to 0.6 V and a step potential of 1.95 mV with a 25 mV amplitude and 0.1 s interval time were applied, recording 615 individual data points for each DPV measurement. Each sample was measured three times.

### 2.3. Statistical Analysis

PCA was performed as an unsupervised tool for dimension reduction. The different e-tongues (Device 1, Device 2, Device 3) were analyzed separately, where data sets for responses from the three working electrodes were merged into one data set of 1845 data points and analyzed together with each repeat measurement and each fluid, with the responses standardized to remove effects of different electrode sizes by having each variable (response from each electrode) scaled to unit variance and mean centered. After preprocessing, PCA of samples was performed using IBM SPSS (IBM Svenska AB, Malmö, Sweden), extracting all factors with an eigenvalue greater than 1, without any additional factor rotation. Furthermore, HCA on the extracted PCs was performed by calculating the Euclidean distance using Ward’s minimum variance method using IBM SPSS.

### 2.4. ^1^H-NMR

Sweat samples collected in conjunction with measurements with the wearable electronic tongue (Device 3) were also prepared for ^1^H nuclear magnetic resonance (NMR) spectroscopy. Fifty µL of sweat was dried under a flow of N_2_ gas to minimize the amount of water in the sample. Afterwards the samples were resuspended/diluted in 350 µL D_2_O. The resuspended samples were transferred into 5 mm NMR tubes. The spectra were collected on a Varian Mercury 400 MHz spectrometer at a resonance frequency of 400.41 MHz using a 5 mm Varian 400 ASW 1H/13C/31P/15N/4NUC PFG 40–162 MHz (SN40P5A910) probe at 25 °C. The spectra were acquired using a 90 °C pulse (12.3 µs pulse width), a relaxation delay of 2 s, an acquisition time of 2.6 s, a spectral width of 6406.1 Hz (−3.3 to 12.7 ppm), with 16,384 complex data points and a 20 Hz spin. The residual water signal was suppressed by using a PRESAT pulse sequence available in VnmrJ version-4.2, using a presaturation delay of 10 s and a power of 36 Hz at 4.65 ppm. All spectra were Fourier transformed using MestReNova (version 14.1.2, Mestrelab Research, Escondido, CA, USA) with zero filling to 64k data points. All spectra were phased and baseline corrected. The area of the peaks was calculated by fitting Lorentzian–Gaussian peaks to the regions of interest.

## 3. Results

### 3.1. Characterization of Electronic Tongue in Complex Buffer and Physiological Fluids

In order to assess the viability of using an e-tongue for classifying different sweat samples, gold, platinum, and palladium electrodes were first investigated in buffer solutions with analytes of interest. Electrode materials for e-tongues have previously been investigated, and the electrode materials were chosen to have different response profiles and provide a simple and robust sensing platform [35]. Nickel and silver electrodes were initially investigated but excluded from the analysis, as they were found to not contribute to the sample separation.

Sweat contains a large number of different compounds which could be of clinical importance, where a few compounds were selected to investigate the response of the electronic tongue [36,37]. It should be noted that very large differences exist in reported concentration ranges in the literature, which makes the choice of analyte concentration for this study somewhat arbitrary. The glucose level in sweat has been shown to correlate with blood, however, the presence of glucose is much lower than in blood, but with very large variations in reported values between different studies (from 0.1 mM up to around 2 mM) [38,39]. Sweat electrolyte concentration can vary widely between individuals, with concentration values of sodium and chloride reported below 10 mM and above 100 mM, with chloride being used for diagnosis of cystic fibrosis [9,10]. Average urea concentrations in sweat range from ~4 to ~12 mM for healthy people, with higher levels in kidney patients [14]. Sweat also contains a variety of proteins, peptides, and amino acids. While only a small proportion of the total protein content in sweat is made up of albumin, albumin was chosen as a model protein to see the effect of a high protein content on the response of the electronic tongue [40]. Sweat also contains different vitamins, where, e.g., loss of vitamin C due to heat exposure is of clinical relevance [41,42]. Finally, lactate and bicarbonate are present in sweat at elevated levels, with concentrations related to the level of physical exertion or sweat rate, but which have also shown clinical relevance in, e.g., cystic fibrosis [43,44]. Some reference values of compounds of interest are summarized in Table 1 below.

As a model experiment, initial investigations were performed to characterize the electrode system in buffer solution, which was also further spiked with either 150 mM sodium chloride, 5 mM glucose, 10 mM lactate, 100 µg mL^−1^ ascorbic acid, 10 mM urea, 100 mM sodium bicarbonate or 10 mg mL^−1^ albumin. Concentrations were chosen at the highest end of the range or even slightly above, in order to also elicit a clear response in the complex background. However, concentrations in the high ranges also typically signal that some physiological issue exists. Typical current responses for the electrodes recorded using DPV are displayed in Figure 1. Different response patterns were obtained for all electrodes. The different electrode materials show non-specific and overlapping signals with different sensitivity properties towards the different analytes. Whereas direct sensing of different analytes in complex fluids would not be possible, the results indicate that by combining the electrodes an e-tongue system could be used to differentiate between samples of different compositions.

When measuring using DPV in complex buffers and physiological fluids, which contain a multitude of different components, the response signal is complex, and a direct interpretation of the data is difficult. Without additional chemical analysis performed on the physiological samples, it is impossible to determine their contents based only on the response of the e-tongue, as the selectivity of individual electrodes is insufficient for specific analysis of single components from such samples consisting of several redox-active compounds and various ions. However, the voltammograms from the e-tongue system contain a large amount of “hidden” information and to extract this information PCA was employed, a useful mathematical tool to explain variance in experimental data, where DPV responses from each electrode were combined into one data-set and used for analysis [47]. Such analysis has the major advantage that no prior knowledge about samples or variables is required and that the data structure is represented by as few variables as possible. The generated score plots show the relation between the experiments, and groupings in the score plots can be used for classification.

PCA was used to characterize the response from the different electrodes in different buffer solutions. Every sample was tested three times using DPV (as shown in Figure 1), and so to perform the analysis of all the samples, the data from each electrode was merged by combining the different data sets to a large data set of size 24 × 1845 (eight different fluids each tested three times, using three different working electrodes each recording 615 data points). Discrimination of the samples was possible by studying the score plots. In total, six different principal components (PCs) were extracted, all with absolute Eigen values larger than one, explaining a total of 94.7% of the variance in the samples, where the first three components (PC 1, PC 2, PC 3) explained 37.2%, 19.1%, and 14.8%, respectively. The first three PCs are illustrated in Figure 2, where the different samples with various added analytes are all well separated. This shows that the e-tongue can be used to distinguish between spiked samples containing different clinically relevant biomarkers that can be present in physiological sweat samples. These results indicate that using an e-tongue for screening of different physiological samples could be a very useful tool to quickly assess the contents of the sample and/or monitoring changes over time, where, e.g., clustering could be used to assess the composition of measured samples.

To investigate the e-tongue for the specific purpose of sweat analysis in more detail, in a similar manner as with the buffer solutions, a sweat sample collected from a healthy volunteer via heat-induced sweating, with unknown content, was also investigated with DPV, with the same analytes as for the investigations in buffer added to the sample. Due to the limited amount of collected sweat, a miniature voltammetric e-tongue was designed by combining micro-wires of gold, platinum, and palladium, with a combined micro-silver counter/reference electrode, to allow electrochemical characterization in real sweat samples of a small volume. Each sample was investigated three times, where the same original sweat sample was used for all the measurements, so as to keep the concentration of different metabolites constant besides that with which the sample was spiked. These measurements were performed to validate that the e-tongue would be able to distinguish different concentrations also when applied to a complex background instead of just buffer solution. In addition to the spiked sweat samples, additional heat-induced sweat samples from three different volunteers were collected and analyzed. All the sweat samples were analyzed together, combining the different data sets to a large data set of size 30 × 1845 (10 different fluids each tested three times, using three different working electrodes each recording 615 data points). The results of the PCA are shown in Figure 3. In total, six different principal components were extracted, all with absolute Eigen values larger than one, explaining a total of 94.6% of the variance in the sample, where the first three components (PC 1, PC 2, PC 3) explained 34.4%, 24.5%, and 18.9%, respectively. Addition of analytes resulted in samples that could be differentiated from each other also when the background solution was real sweat instead of PB. Furthermore, the samples collected from different volunteers gave a widely different response, related to a large individual variation in the composition of the sweat, and all are clearly separated in the PCA score plots. However, the exact composition of each fluid was unknown.

Sweat is a very complex fluid and a detailed appreciation of different components is not possible due to the large possible variability in content. To fully understand the cause of the sample discrimination, extensive reference measurements would be required. As all the samples have a different chemical composition, possible explanations of the various response patterns are electrode kinetics, metal oxide-catalyzed reactions, and adsorption of different species present in the solution. However, the aim of the experiments was not to develop a complete analytical system, but to investigate the possibilities of using DPV combined with multivariate methods for classification purposes for non-invasive analysis. Nonetheless, these results demonstrate that an e-tongue can be used to distinguish between real sweat samples of different composition.

To further investigate the possibility of using the e-tongue in sweat, the stability of the sensor response was investigated in pre-collected sweat. The sensors were placed in the sweat solutions and DPV was recorded right away, after 0.5 h, 2 h, and 8 h in the same solution. Finally, the sensors were removed, cleaned by the standard procedure then placed back in the sweat solution and measured one final time. The results are shown in Figure 4. Overall, the different sensors gave a similar response over time, especially over shorter measurement times. After long incubation in sweat larger changes were observed. The changed response can be attributed both to a change of the sensor surface, such as adsorption of different species changing the redox properties, as well as the changing composition of the sweat solution itself. It is likely that a collected sweat solution contains different cells and bacteria, which over time can influence the redox response of the sensors as well as change the composition of the sweat. After cleaning, part of the signal is restored but some permanent changes remain, which can be attributed to compositional changes in the sweat such as e.g., bacterial growth. If worn on the skin, it is likely this problem can be lessened by carefully cleaning the sensor site before attachment.

### 3.2. Wearable E-Tongue for On-Body Sweat Characterization

As shown in Section 3.1, a voltammetric e-tongue can be used to separate complex physiological fluids of different compositions, which could be used for example to determine attributes of the fluids or to monitor changes in the fluids. Sweat is of particular interest as it can be monitored non-invasively, contains a wealth of physiologically and metabolically important biomarkers, and shows a large variability in composition [5,36,39,48]. To allow for sweat monitoring directly on the skin, a wearable e-tongue prototype was designed, shown in Figure 5. Au, Pt, and Pd were plated on a flexible circuit board and used as working electrodes, with an Ag counter/reference electrode.

Prior to on-skin measurement, the wearable e-tongue was characterized with a small (roughly 20 µL per measurement) amount of pre-collected sweat, spiked with the same concentration of analytes as described above (in Section 3.1). As the actual composition of all sweat samples is unknown apart from the analyte it was spiked with, these pre-body measurements are important in order to be able to interpret changes in the on-body measurements and be able to relate them to possible compositional changes of the analyzed fluid. The e-tongue prototype was then attached to the back of a healthy volunteer whereupon a 5-min high-intensity exercise regime was performed, triggering heavy sweating. A lighter exercise load was then maintained throughout the measurements, to keep an elevated level of perspiration. Directly after finishing the 5-min exercise, the e-tongue was connected to a multichannel potentiostat and measurements were performed with the wearable e-tongue attached to the back of the volunteer. After an additional 15-min period, new measurements were performed, in order to study the changing composition of the exuded sweat. All the data using the wearable e-tongue were combined and analyzed together, combining the different data sets to a large data set of size 27 × 1845 (nine different fluids each tested three times, using three different working electrodes each recording 615 data points).

As shown in Figure 6, after analysis of the data, PCA explained a total of 87.8% of the variance, where five PCs were extracted, all with absolute Eigen values larger than one. The PCs explained a total 47.4%, 21.7%, 8.3%, 6.2%, and 4.2% of the variance, respectively. A detailed breakdown of one of each repeat measurement of different fluids per PC is also listed in Table 2. To further differentiate between the samples based on the five PCs, HCA was performed by calculating the Euclidean distance using Ward’s minimum variance method. The distances between each of the first measurements in the different sample fluids are shown in Table 3, which gives information about the overall similarity of the different samples taking into account all five PCs. The results shown in Figure 6 show overlap between some of the different samples, to a larger extent than the miniature e-tongue (Figure 3). However, taking into account all five PCs, sample differentiation is possible. For example, ascorbic acid show overlap on the first three PCs, but is separable from the other analytes on PC 4 and PC 5. A notable difference is observed between the in vitro sweat measurements and on-body sweat measurements, where the on-body measurements are clearly separated from the pre-body measurements on the first two PCs which explain most of the sample variance (Figure 6a). This is not surprising, as different sources of sweat were used for the measurements, where the in vitro analysis used pooled sweat collected via heat-induced sweating and the on-body sweating was caused by an exercise regime. It is well-known that different sources of sweat can vary widely in composition [49,50]. The response of the wearable e-tongue on the skin directly after exercise and with a 15 min delay was quite similar, with the largest differentiation being the samples recorded on the loading of PC 3, which decreased from around 0.22 to −0.61. To assess the possible compositional changes in the sweat samples recorded in connection with exercise, the results were compared with the results from the in vitro measurements, where samples were spiked with a known analyte. As shown in Figure 6 and Table 2, a large decrease in PC3 was observed when the sample was spiked with lactate, changing the loading from around 0.4 to −0.15. These results indicate that an increasing lactate concentration in the delayed measurement on the skin compared to the measurement taken directly after the 5-min exercise could be one of the causes for the sample separation. Based on the distance between the samples shown in Table 3, it also can be observed that the last on-body measurement is closer to the sweat sample spiked with lactate than the first on-body measurement, with distances of 2.728 and 3.533 versus lactate, respectively. Whereas other compositional changes surely existed between the samples, these are difficult to assess due to the complexity of sweat composition.

### 3.3. NMR-Analysis of Sweat Samples Characterized by the Wearable E-Tongue

In order to assess if the indicated changes detected by the e-tongue corresponded to actual changes in the measured samples, sweat samples were collected from the back in connection with the e-tongue measurements, both directly after the exercise program and 15 min later. The collected samples were then analyzed using ^1^H NMR spectroscopy. This was done to assess compositional changes that could lead to the different responses generated by the electronic tongue. Several different studies have been aimed at characterizing the composition of sweat, using techniques such as mass spectroscopy and NMR. The studies have shown sweat to display a large physiological variability in terms of the number of metabolites present and the concentrations between subjects as well as between different body locations [49,50,51,52].

^1^H NMR spectroscopy produced spectra of the sweat samples that identified and quantified different metabolites, as shown in Figure 7 with identified compounds listed in Table 4. The metabolites were identified by comparison to previously published data on common endogenous metabolites [53,54,55]. Several peaks could not be identified, as verification of the identities of these metabolites would require the extensive use of either, e.g., 2D NMR spectroscopy or high-performance liquid chromatography coupled to mass spectroscopy (HPLC-MS). In general, the initial sweat sample contained a myriad of peaks, most of which were reduced in intensity or completely lacking in the sample taken after 15 min. This is not surprising, as the initial sweat sample would contain partially old sweat and contamination from the skin surface, where the contamination effect of all these sources will be diluted and diminished when higher sweat rates are maintained over time [5,6]. Lactate was the most dominant metabolite found in both samples and also displayed the largest change between samples, where the amount increased roughly six-fold in the sample taken 15 min after the exercise program. Another significant metabolite was glycerol, which had similar concentrations before and after the exercise program. These results agree with the analysis using the wearable e-tongue, where the decrease in PC 3 could be attributed to an increase in lactate concentration. The results also makes sense from a physiological perspective, as high exertion exercise regimes increase the metabolic activity of the sweat gland itself and have been shown to display a spike in lactate correlating with exercise intensity (sweat rate), whereas this does not necessarily represent the anaerobic state of the body [6].

## 4. Discussion

Previous studies of sweat have focused on employing specific sensors, where enzymatic sensors for sweat biosensing demonstrations (e.g., ethanol, lactate, glucose) and ion-selective electrodes are common techniques [11]. Here, we take instead a new approach of sweat characterization, by combining several non-specific electrodes, which gives an electrochemical profile of the sample analyzed, which changes with changing analyte concentrations. The results of the wearable e-tongue in combination with the NMR analysis of the measured sweat show the potential of using an e-tongue for non-invasive analysis of sweat samples. To improve the performance of the wearable e-tongue, more electrodes could be included for the DPV measurements, e.g., other noble metals such as iridium or rhodium. Electrodes could also be further modified with, e.g., self-assembled monolayers, conducting polymers or redox catalysts to differentiate the response. Other types of measurements could also be included in the analysis, e.g., inclusion of ion-selective electrodes and conductivity measurements. In addition, while DPV is a very sensitive technique, due to the complex composition of sweat and the typically very low amounts of some analytes, e.g., glucose which typically vary in the micro-molar range, minor concentration changes would not be detectable. The e-tongue could however also be complemented with specific sensors for analytes present at very low concentrations, with the added benefit that the data analysis of the combined sensor array would be improved. The combination of different sensor technologies in hybrid electronic tongues has been shown to improve the performance [21,27].

In addition to incorporating additional sensor modalities, several other design considerations should be considered. First, the volume of sweat required to operate the device could be an issue, as sweat generation rate varies from 0.1 nL min^−1^ per gland to >10 nL min^−1^ per gland and eccrine sweat gland densities range from tens to hundreds of glands per square centimeter depending on body location [5,6]. For a wearable e-tongue to be viable for applications outside of active perspirers (such as athletes or very active people), artificial sweat stimulation is required via, e.g., iontophoretic delivery of pilocarpine [8,11]. Second, sensor attachment to the skin is an important consideration. Body movement can cause the sweat flow to accelerate or even reverse and damage or alter the sensor itself with pressure or abrasion. In our developed wearable e-tongue, we used a stretch band to attach the sensor, which was fine for our purpose of sensing during active perspiration, when sweat flows easily, but otherwise not ideal. Possible solutions could be to deploy a wicking system to transport the sweat to the sensor or a closed channel system [56,57]. Alternatively, a flexible tattoo-based sensing system could be employed, combined with a localized iontophoretic sweat stimulation system [46]. Finally, care should be taken regarding the reference electrode, and as we describe here a combined counter/reference electrode is used. Operation of the sensor can lead to a significant potential shift of the reference, causing a significant reduction in function of the sensing system [34]. For the wearable e-tongue used in this study, employing three working electrodes with a combined counter/reference electrode nine times the size of the working electrode, is not an issue, but by adding additional working electrodes of other materials or other sensor modalities this could become an issue and should be taken into account in the design of the reference electrode, where a more complicated three-electrode system may be more suitable.

By extending the number of volunteers used in the study and combining the response of the e-tongue with careful chemical analysis of sweat samples, powerful machine learning algorithms could be deployed to achieve better sample classification and characterization. However, this requires a very large number of volunteers to be recruited and investigated. As the chemical composition of perspiration varies between individuals, sample location on the body and the causes and period of sweating, with other factors, e.g., having a fever also influencing its composition, carful studies with regard to sampling, sensor placement, etc. enrolling a large number of volunteers would be needed [58]. By showing here the important first step, that the wearable e-tongue indeed can separate samples of changing composition, also verified with NMR, we intend to improve in future studies the original design according to the above discussed points to be deployed in a large-scale study. Sensors and algorithms for analysis of the data provided by such a wearable sensor array could then be implemented in, e.g., a smart watch, to provide insight into physiological changes.

## 5. Conclusions

An electronic tongue was designed by combining different metal electrodes as working electrodes and showed promising ability when applied in complex fluids, such as different buffer solutions and sweat. The different samples could be discriminated using principal component analysis. Based on these results, a wearable prototype electronic tongue was designed and used to differentiate between the changing composition of the exuded sweat upon exercise over time. The results showed great promise in differentiating sweat samples of different composition. When combined with ^1^H-NMR analysis of the sweat samples, some of the observed changes measured with the e-tongue could be attributed to the changing lactate concentrations in samples after physical exercise.

## Figures and Tables

**Figure 1 sensors-21-07311-f001:**
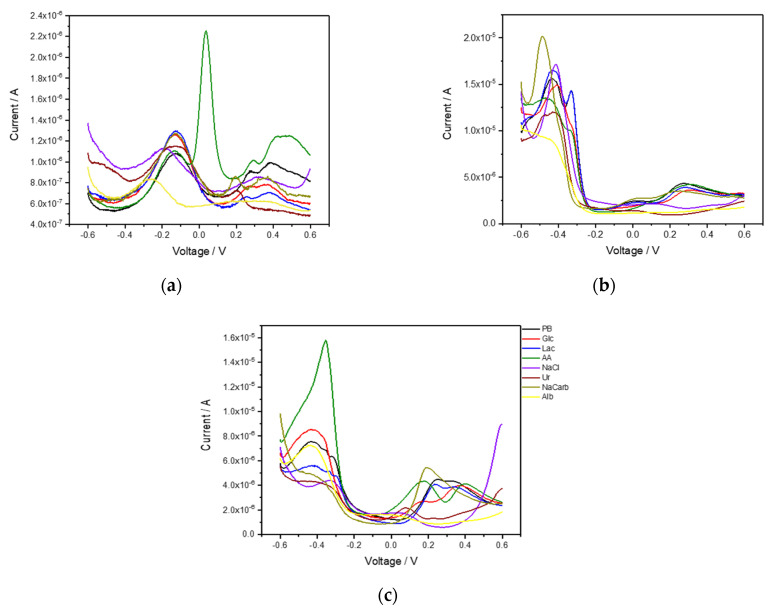
Differential pulse voltammetry (scan rate 20 mV/s, pulse amplitude 25 mV, pulse time 50 ms) of (**a**) gold, (**b**) platinum and (**c**) palladium electrodes in PB and PB spiked with 5 mM glucose (Glc), 10 mM lactate (Lac), 100 µg mL^−1^ ascorbic acid (AA), 150 mM sodium chloride, 10 mM urea (Ur), 100 mM sodium bicarbonate (NaCarb) or 10 mg mL^−1^ albumin (Alb).

**Figure 2 sensors-21-07311-f002:**
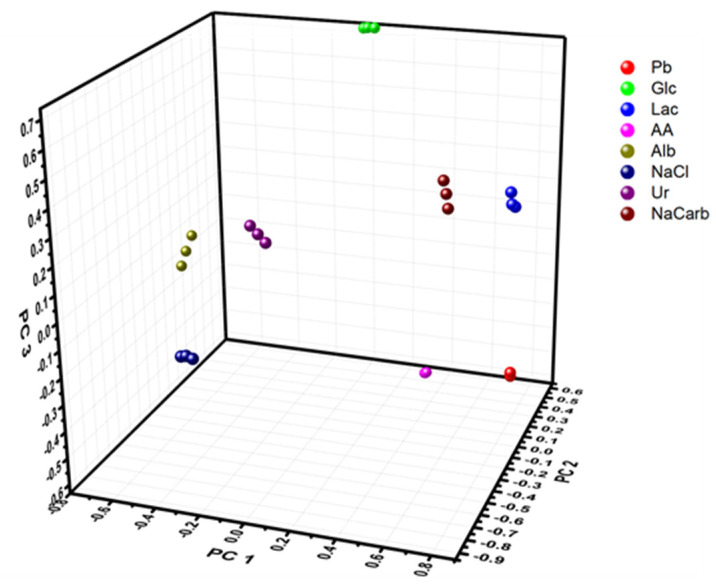
PCA score plot of three different repeat measurements using a macro e-tongue (Device 1) in PB spiked with different metabolites; 5 mM glucose (Glc), 10 mM lactate (Lac), 100 µg ml^−1^ ascorbic acid (AA), 10 mg mL^−1^ albumin (Alb), 150 mM sodium chloride, 10 mM urea (Ur) or 100 mM sodium bicarbonate (NaCarb). Explained variance of PC 1 (37.2%), PC 2 (19.1%), and PC 3 (14.8%).

**Figure 3 sensors-21-07311-f003:**
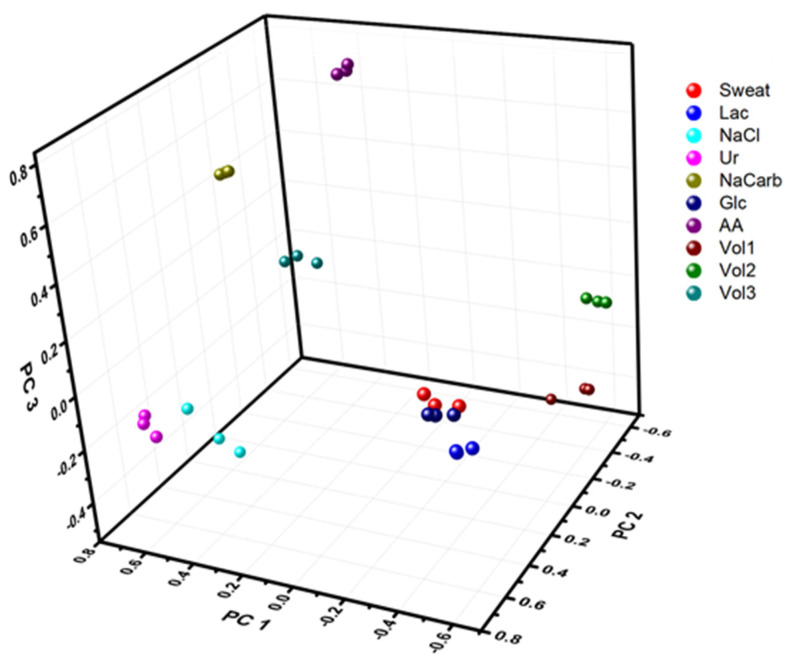
PCA score plot of three different repeat measurements using a micro e-tongue (Device 2) in sweat samples collected from three different volunteers (Vol 1, Vol 2, Vol 3) and sweat spiked with different metabolites; 5 mM glucose (Glc), 10 mM lactate (Lac), 100 µg mL^−1^ ascorbic acid (AA), 10 mg mL^−1^ albumin (Alb), 150 mM sodium chloride, 10 mM urea (Ur) or 100 mM sodium bicarbonate (NaCarb). Explained variance of PC 1 (34.4%), PC 2 (24.5%), and PC 3 (18.9%).

**Figure 4 sensors-21-07311-f004:**
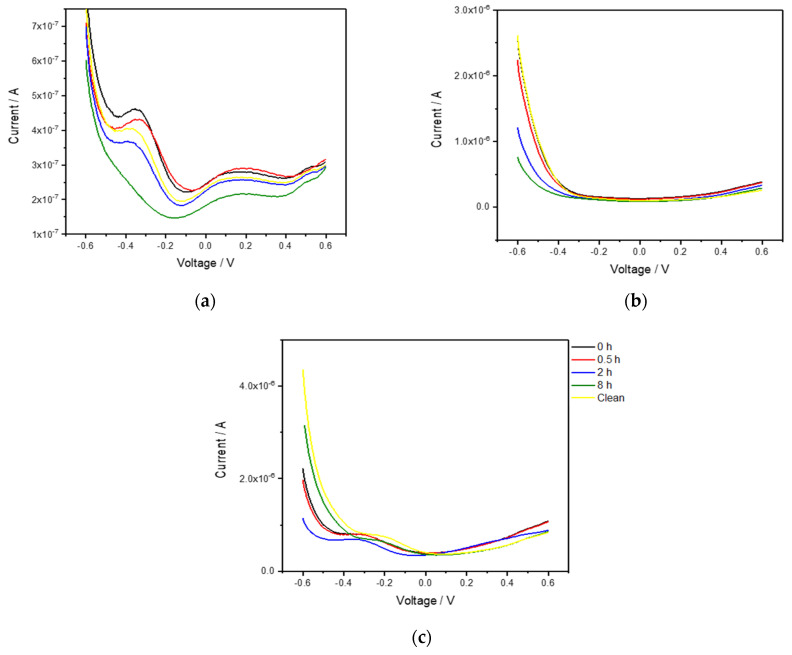
Differential pulse voltammetry (scan rate 20 mV/s, pulse amplitude 25 mV, pulse time 50 ms) of (**a**) gold, (**b**) platinum, and (**c**) palladium electrodes incubated in the same sweat solution with measurements taken immediately, after 0.5 h, 2 h, 8 h and finally after cleaning the electrodes and measuring again in the same solution.

**Figure 5 sensors-21-07311-f005:**
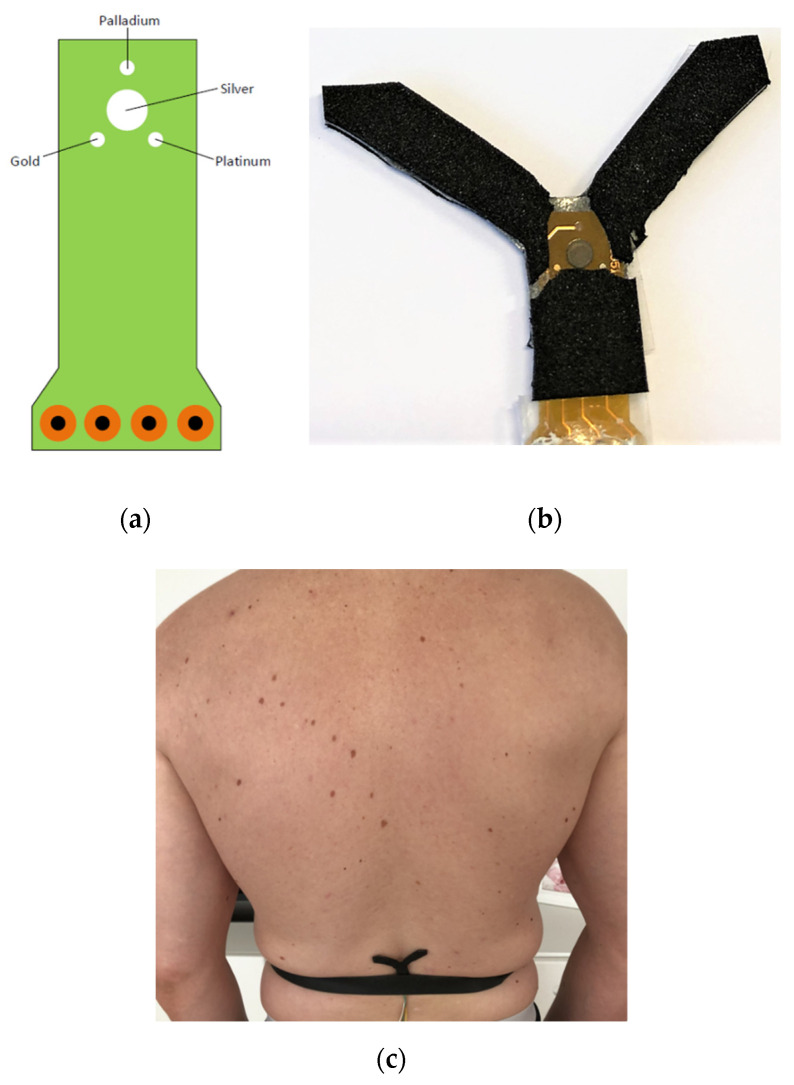
(**a**) Sketch showing the general design of the flexible circuit board. (**b**) Photograph of a wearable e-tongue prototype. (**c**) Wearable e-tongue attached to volunteers back, worn during exercise and used for on-body analysis of the subject’s sweat.

**Figure 6 sensors-21-07311-f006:**
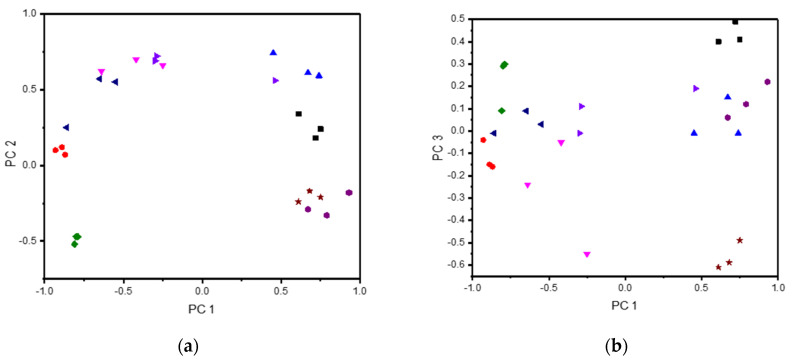
PCA score plots of three different repeat measurements using a wearable e-tongue (Device 3) in vitro in sweat spiked with different metabolites and on body directly after (SweatFirst) and 15 min after (SweatLast) an exercise regime. (**a**) PC 1 vs PC 2. (**b**) PC 1 vs PC 3. (**c**) PC 2 vs PC 3. (**d**) PC 4 vs PC 5. Explained variance of PC 1 (47.4%), PC 2 (21.7%), PC 3 (8.3%), PC 4 (6.2%), and PC 5 (4.2%).

**Figure 7 sensors-21-07311-f007:**
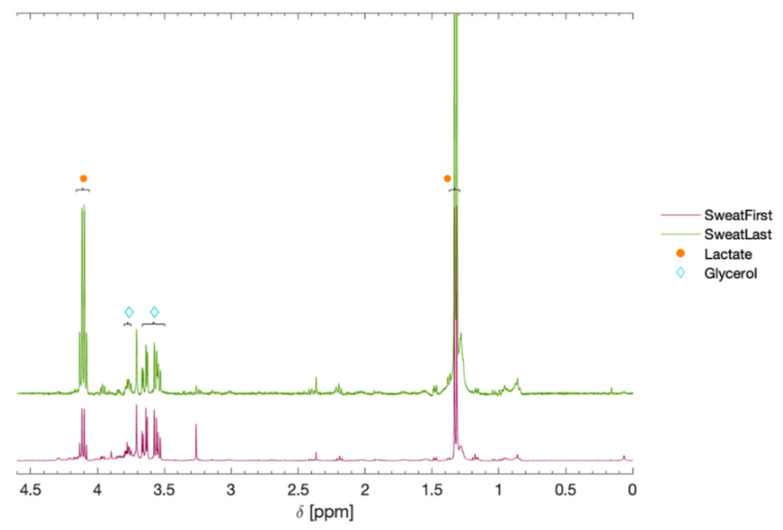
The 400 MHz ^1^H-NMR spectra of human sweat taken directly after (SweatFirst), as well as after 15 min (SweatLast), following a 5 min exercise regime. The spectra show a significant difference for the lactate (orange circles) in the two samples, but also a distinct presence of glycerol (turquoise diamonds).

**Table 1 sensors-21-07311-t001:** Reference values for compounds of interest present in sweat.

Compound	Concentration (mM)
Lactate	5–40 (Ref [45])
Glucose	0.1–2 (Ref [38,39])
Sodium carbonate	0.5–100 (Ref [36,43])
Na^+^, Cl^−^	below 10, above 100 (Ref [9,10])
Urea	5– above 100 (Ref [14,45])
Ascorbic acid	0.01–0.5 (Ref [45,46])

**Table 2 sensors-21-07311-t002:** List of PC score values for the first of each repeat measurement of sweat, sweat spiked with different compounds, and on-body measurements, obtained from the wearable e-tongue.

Compound	PC 1	PC 2	PC 3	PC 4	PC 5
Sweat	0.61	0.34	0.40	−0.44	0.03
Lac	−0.89	0.12	−0.15	−0.01	−0.09
NaCarb	−0.81	−0.52	0.09	0.10	−0.06
Glc	−0.86	0.25	−0.01	0.11	0.06
Ur	−0.42	0.70	−0.05	−0.35	−0.02
AA	−0.30	0.69	−0.01	0.37	0.40
NaCl	0.45	0.74	−0.01	0.19	−0.27
SweatFirst	0.93	−0.18	0.22	0.4	−0.07
SweatLast	0.61	−0.24	−0.61	-0.3	0.28

**Table 3 sensors-21-07311-t003:** Euclidean distance between each of the first repeat measurements of the different samples, based on the 5 extracted PCs using the results obtained from the wearable e-tongue.

Sample	Sweat	Lac	NaCarb	Glc	Ur	AA	NaCl	SweatFirst	SweatLast
Sweat	0	2.823	3.163	2.643	1.409	1.901	0.846	0.632	1.602
Lac		0	0.492	0.393	0.687	1.184	2.278	3.533	2.728
NaCarb			0	1.242	1.862	2.003	3.230	3.142	2.680
Glc				0	0.614	0.684	2.070	3.430	2.805
Ur					0	0.702	1.117	2.792	2.457
AA						0	1.039	2.611	2.183
NaCl							0	1.176	1.673
SweatFirst								0	0.902
SweatLast									0

**Table 4 sensors-21-07311-t004:** Peaks assigned to metabolites in sweat samples, collected in conjunction with measurements with the wearable e-tongue, after comparison with an NMR spectroscopic metabolite reference database. Multiplicity is indicated as s = singlet, d = doublet, t = triplet, q = quartet, m = multiplet. The relative area is calculated per molecule.

Metabolite	Chemical Shift (δ)	Area, First	Area, Last
Lactate	4.11 (q), 1.32 (d)	1.0	5.9
Serine	3.96 (m), 3.85 (m)	trace	trace
Alanine	3.79 (q), 1.48 (d)	trace	trace
Glycerol	3.77 (m), 3.65 (m), 3.55 (m)	0.5	0.6
Pyruvate	2.37 (s)	trace	trace
Lipids (-CH_2_-)	1.28 (broad)	trace	trace
Lipids/Protein (CH_3_-)	1.15 (broad)	trace	trace

## Data Availability

Most data is contained within the article. Additional data presented in this study are available on request from the corresponding author, due to containing data from human volunteers.

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
