# Peer review of "Wearable Electronic Tongue for Non-Invasive Assessment of Human Sweat"

_sensors, 2021, doi:10.3390/s21217311_

Round 1
Reviewer 1 Report
In this manuscript, the authors presented an e-tongue approach to differentiate sweat contents. By sing metal electrodes with electrochemical analyses, the authors first calibrate the signals from different electrodes for various bio-chemical substances. A score system was also developed to provide qualitative examination for sweat. A wearable seat collector was also fabricated to collect fresh sweat samples from human body. The sweat was then analyzed to show the feasibility of the developed algorithm. Overall, the manuscript is written in a rational organization with good experiment designs. But certain details were not well described for the readers to repeat their findings. It is thus recommended for a minor revision. My comments are listed below for the authors.
- On page 4, line 163-180, the authors mentioned the detection requirements for various species. But the contents are a bit scattered and not easy to understand. It might be better to summarize them into a table to improve the readability.
- The algorithm for PCA in line 214-230 was not well clarified. Please provide details for the calculation of scores in Figure 2.
- How does the PCA scores in the tests associated with weight fractions or concentrations for each species in the samples? The NMR results seemed to only provide qualitative comparison but quantitative results are necessary to for the e-tongue.
- The e-tongue needs to be operated in sweat for at least few hours. Some stability tests are needed to check the consistency of the signals after long-term operation.
Reviewer 2 Report
Revised manuscript reports a wearable array of voltammetric sensors, formed by different metal electrodes, intended to perform chemical analysis of different chemicals in human sweat. Nothing in connection with a 3 noble metal array plus use of PCA data treatment suggest any novelty in the field of electronic tongues. Nevertheless, implementation of these in a wearable device, and verification of its ability to discriminate a number of chemicals in real sweat samples are worth considering in the field.
A number of observations arised during examination of the manuscript that need consideration by the authors:
1. Abstract starts too discursively, and later missed to communicate most of the details of the work. Delete first two sentences, and add transduction used (voltammetric), comment what technology is used to make the device wearable (flexible circuit board), specify the multivariate data treatment employed. This information deserves also being communicated on last paragraph of introduction (Pag. 2).
2. Introduction. There are significant works in the literature, connected with diagnosis of cystic fibrosis, that did sweat analysis with an array of potentiometric sensors (Analyst 125 (2000) 2264). Please cite.
3. Pag 2, line 76. Cite the multiple entries as [20-28].
4. Materials and methods. Add subsections to differentiate chemicals, apparatus, sensor devices, e-tongue procedure.
5. Materials and methods. As I understand, 3 electronic tongue set-ups are used along this study. This could be made clearer, specifying Device 1, Device 2 and Device 3. The micro-etongue should be better described, featuring the dimensions of the device (and/or the volume of the electrochemical cell).
6. Pag 3, line 100. “ … the prototype was designed by plating gold, palladium and platinum …”, plating onto what? Cu discs on PCB? From what solutions? With what procedural details?
7. Materials and methods. How did the authors chose the concentrations of the different chemicals studied? It would be interesting to relate these values with normal, rest condition, physiological values?
8. Pag 3, line 105. If Figure 4b is the first figure cited, it should be Figure 1, and be placed close to this point.
9. An important issue for sensing devices is their figure of reproducibility. This is what can guarantee their use for a given time, something important specially in electronic tongue systems, that normally employ big effort for its calibration. This is well described in specific reports in the literature, which I recommend the authors to follow (Trends in Anal Chem 121 (2019) 115675).
10. Figures 2 and 3. Reduce displayed info, as these are only preliminary experiments. Display only PC1 and PC2 score plots or provide a 3D plot.
11. Table 1. Values stated here must be PC score average values; PC loadings are a different matter here.
12. Section with the NMR experiments seem too inconclusive. Only lactate gave any appreciable result. I suggest removing these preliminary experiments from the manuscript (or moving to a supplementary information file).
13. Page 13, lines 393-394. Many possibilities are available if incorporating modified sensors, e.g. redox catalysts, metal nanoparticles, conducting polymers, or others, in the array.
14. Page 13, lines 414-415. There are other attempts in the literature that perhaps should be mentioned, like those using adhesive or tattoo concepts.
15. Ref 18 is not a good example. By definition, a single electrochemical sensor cannot constitute and electronic tongue.
16. Ref 29. The journal name in this reference is missing.
Round 2
Reviewer 2 Report
Authors have conducted an extensive revision, contemplating all the comments supplied. Being this satisfactory, the manucript may be published as it is now.